

# Salinity changes and anoxia resulting from enhanced runoff during the late Permian global warming and mass extinction event

Elsbeth E. van Soelen[1], Richard .J Twitchett[2], Wolfram M. Kürschner[3]

[1]University of Oslo, Departments of Geosciences, P.O box 1047 Blindern 0316 Oslo, Norway

[2]Natural History Museum, Earth Sciences Department, London, SW7 5BD, UK

[3]Department of Geosciences and Centre for Earth Evolution and Dynamics (CEED), University of Oslo, Oslo, Norway

*Correspondence to:* Elsbeth E. van Soelen (e.e.v.soelen@geo.uio.no)



**Abstract.** The Late Permian biotic crisis had a major impact on marine and terrestrial environments. Rising $CO_2$
levels following Siberian Trap volcanic activity were likely responsible for expanding marine anoxia and
elevated water temperatures. This study focusses on one of the stratigraphically most expanded Permian-
Triassic records known, from Jameson land, east Greenland. High resolution sampling allows for a detailed
reconstruction of the changing environmental conditions during the extinction event and the development of
anoxic water conditions. Since very little is known about how salinity was affected during the extinction event,
we especially focus on the aquatic palynomorphs and infer changes in salinity from changes in the assemblage
and morphology. The extinction event, here defined by a peak in spore/pollen, indicating disturbance and
vegetation destruction in the terrestrial environment, postdates a negative excursion in the total organic carbon,
but predates the development of anoxia in the basin. Based on the newest estimations for sedimentation rates,
the marine and terrestrial ecosystem collapse took between 1.6 to 8 kyrs, a much shorter interval than previously
estimated. The palynofacies and palynomorph records show that the environmental changes can be explained by
enhanced runoff, increased primary productivity and water column stratification. A lowering in salinity is
supported by changes in the acritarch morphology. The length of the processes of the acritarchs becomes shorter
during the extinction event and we propose that these changes are evidence for a reduction in salinity in the
shallow marine setting of the study site. This inference is supported by changes in acritarch distribution, which
suggest a change in palaeoenvironment from open marine conditions before the start of the extinction event to
more near-shore conditions during and after the crisis. In a period of sea-level rise, such a reduction in salinity
can only be explained by increased runoff. High amounts of both terrestrial and marine organic fragments in the
first anoxic layers suggest that high runoff, increased nutrient availability, possibly in combination with soil
erosion, are responsible for the development of anoxia in the basin. Enhanced runoff could result from changes
in the hydrological cycle during the late Permian extinction event, which is a likely consequence of global
warming. In addition, vegetation destruction and soil erosion may also have resulted in enhanced runoff.
Salinity stratification could potentially explain the development of anoxia in other shallow marine sites. The
input of fresh water and related changes in coastal salinity could also have implications for the interpretation of
oxygen isotope records and sea water temperature reconstructions in some sites.



## 1. Introduction

The late Permian extinction event was the most severe global crisis of the Phanerozoic in terms of both taxonomic loss and ecological impact (e.g. McGhee et al., 2012). The current consensus is that the extinction was likely due to global warming and associated environmental changes caused by $CO_2$ emissions from Siberian Trap volcanic activity, because of the close timing between the volcanic activity and the extinction event (e.g., Burgess et al., 2017; Burgess and Bowring, 2015). Most studies on the late Permian extinction have inferred that expanding marine anoxia (e.g. Wignall and Hallam, 1992) is a key biotic factor causing marine extinction and ecosystem collapse. There are different theories to explain the spreading of anoxia, which affected both deep and shallow sites (e.g. Bond and Wignall, 2010; Isozaki, 1997; Wignall and Twitchett, 1996). A weakened temperature gradient between equator and pole would have slowed ocean circulation and may have facilitated expansion of the oxygen minimum zone (Hotinski et al., 2001). Increased weathering and detrital input (Algeo and Twitchett, 2010), and soil erosion (Sephton et al., 2005) led to enhanced terrestrial matter input in marine sections, and may also have contributed to eutrophication and in stratified waters led to hypoxia or anoxia (Sephton et al., 2005). Other factors that are thought to play important roles in the extinction are elevated water temperatures (e.g. Sun et al., 2012), and ocean acidification (Clapham and Payne, 2011). One important environmental parameter that has received relatively little attention is salinity, even though low salinity ocean conditions were once considered to be a leading cause of the marine extinction (Fischer, 1964; Stevens, 1977). Furthermore, potential impacts of changes in salinity, which might be expected from enhanced discharge of freshwater into shelf seas (Winguth and Winguth, 2012), have been largely ignored.

Microfossils of marine algae are excellent recorders of environmental change in the water column. Of these, the organic-walled cysts of dinoflagellates have proven especially useful in palaeo-environmental studies (e.g. Ellegaard, 2000; Mertens et al., 2009, 2012, Mudie et al., 2001, 2002; Sluijs and Brinkhuis, 2009; Vernal et al., 2000). While dinocysts are absent or sparse in Palaeozoic deposits, acritarchs are commonly recorded (e.g. Tappan and Loeblich, 1973), even during the late Permian when acritarch diversity was declining (Lei et al., 2013b). Acritarchs are a group of microfossils of organic composition and unknown affinity (Evitt, 1963). Many acritarchs are, however, found exclusively in marine rocks, and some are thought to be precursors of modern dinoflagellate cysts (e.g. Servais et al., 2004). Several studies in East Greenland have reported fluctuating abundances of acritarchs during the Late Permian and Early Triassic (e.g. Balme, 1979; Piasecki, 1984; Stemmerik et al., 2001), but very few late Permian studies have documented the relative abundance of the different genera of aquatic palynomorphs (for example, Shen et al., 2013). Similar to dinocysts, acritarch morphology has been linked to environmental conditions, and acritarchs with longer processes are generally more abundant in more open marine settings (Lei et al., 2012; Stricanne et al., 2004). Both dinocyst and acritarch studies show that salinity is an important, albeit not the only, factor that influences cyst morphology. It is thought that the longer processes in higher salinities stimulate clustering with other cysts or particles in the water column, and thus enhance sinking to the seafloor (Mertens et al., 2009).

The rock record of Jameson Land, East Greenland, has provided key insights into marine environmental changes during the late Permian extinction event. Previous work on this section by Twitchett et al. (2001) showed that collapse of marine and terrestrial ecosystems was synchronous and took between 10 to 60 kyr. Palynological work by Looy et al. (2001) showed that the terrestrial ecosystem collapse is characterized by a





distinct rise in spores, indicative of the loss of woodland and increase in disturbance taxa. The onset of anoxic
conditions is not as abrupt at this location as seen in some other sections, but instead Wignall and Twitchett,
(2002) found unusual alternating patterns of bioturbated and laminated siltstones in the top of the Schuchert Dal
Formation and lowest of the Wordie Creek Formation (Fig. 1). In a recent study, Mettam et al., (2017) showed
that rapidly fluctuating redox conditions occurred during the late Permian extinction, while sedimentological
observations imply that sea level was rising in this period. Estimations of sedimentation rates indicate that each
bioturbated/laminated rock interval, which are between 2-10 cm thick, have been deposited in a period of 50-
1000 years (Mettam et al., 2017). To study the environmental conditions associated with the deposition of these
sediments we look in detail at a short (9 m) interval covering the late Permian extinction and the Formation
boundary between the Schuchert Dal and Wordie Creek Formation. Palaeoenvironment is studied by looking at
variations in the palynofacies (organic particles) and the acritarch assemblage. In addition we present a record of
morphological changes within the acritarch *Micrhystridium*. Runoff and erosion, salinity, sea-level fluctuations
and temperature rise are discussed as possible reasons for the changing environmental conditions.

## 2.    Geological setting and stratigraphy

Samples were collected at the Fiskegrav location of Stemmerik et al. (2001), an outcrop in East Greenland
(N71°32'01.6", W024°20'03.0"), in a small stream section on the east side of Schuchert Dal (Fig. 1). The
section was deposited within a relatively narrow, north-south oriented basin (Stemmerik et al., 2001; Wignall
and Twitchett, 2002). Active rifting and rapid subsidence has resulted in the deposition of one of the most
expanded P-Tr sections. At the Fiskegrav location there are no obvious breaks in sedimentation, even though
large erosive channels exist in P-Tr sections in more northern locations (Twitchett et al., 2001; Wignall and
Twitchett, 2002).

102           The Fiskegrav section shows a transition from the Schuchert Dal Formation into the Wordie Creek

Formation. The Permian/Triassic boundary, defined by the first occurrence of the conodont *Hindeodus parvus*
(Yin, 1996), is located at 23.5 m above the base of the Wordie Creek Formation (Twitchett et al., 2001). This
study focusses on a ca. 9 m section interval including the late Permian extinction and the formation boundary
(Fig. 1). The upper part of the Schuchert Dal Formation consists of blocky (bioturbated), micaceous, greenish
mudstones (muddy siltstones), whereas the lower Wordie Creek Formation consists mainly of laminated, dark
grey mudstones (clay-siltstones) that often contain framboidal pyrite (Twitchett et al., 2001; Wignall and
Twitchett, 2002). The extinction event occurs in the upper metres of the Schuchert Dal Formation, and the start
of the biotic crisis is synchronous in marine and terrestrial ecosystems (Looy et al., 2001; Twitchett et al., 2001).
For more details on biostratigraphy see Mettam et al., (2017); Stemmerik et al., (2001) and Twitchett et al.,

112   (2001).


## 3. Material and Method

### 3.1 Material



A total of 36 samples were collected from the top 4.5m of the Schuchert Dal Formation and the lower 4.2m of
the Wordie Creek Formation. Sample resolution is highest (10cm intervals) in the 2.5m interval covering the
formation boundary and the extinction event. Depths are given in m, relative to the formation boundary between
Schuchert Dal and Wordie Creek Fm. Some of the samples used in this study were also analysed by Mettam et
al. (2017) during their study of changes in redox conditions at the formation boundary.

**3.2 palynofacies and palynological analysis**
Samples (of ca.5 g) were crushed, and a tablet with a known amount of lycopodium spores was added
to allow quantification of organic particles and palynomorphs. Samples were then treated with hydrochloric acid
and hydrofluoric acid to remove carbonates and silicates, and subsequently sieved through 7 µm sieves. Heavy
liquid separation was used to remove heavy minerals like pyrite. The residue, containing palynomorphs and
other organic particles, was mounted onto microscope slides. Material was counted using a Leitz Diaplan
microscope and an AxioCam ERc 5s camera attached to a computer with Zen microscope software (Zen 2 lite,
2011). A minimum of 300 particles per sample were counted for palynofacies analyses. Aquatic palynomorphs
were counted to reach 100 specimens when abundant and at least 30 specimens when abundance was low. Still,
due to low abundances during the biotic crisis, in some samples counts were below 30 (these 6 samples are
indicated in Fig. 4).

**3.3 Acritarch measurements**
The genus *Micrhystridium* includes many species, of which ca. 20 are known from the Permian-
Triassic interval, which differ in their body size, number of processes and process length (Lei et al., 2013b;
Sarjeant, 1970).  Only those with a spherical central body and many (>10) simple processes with closed tips,
were included in this study (i.e. the *M. breve* Group, according to Lei et al., 2013b).  Thus, amongst others, *M.*
*pentagonale* was excluded. Specimens were excluded if they were too damaged (folded or broken) to allow
measurements. To counteract the effects of compression (which can make the body look longer), body size was
calculated from the average of two linear measurements perpendicular to each other. For each specimen three
processes were measured (all within one focal plane). The aim was to measure 20-30 specimens per sample, but
acritarch concentrations were low during the biotic crisis interval, and therefore 5 samples had lower counts
(indicated in Fig. 6).

**4. Results**
**4.1 Palynofacies**
The most abundant organic particles are phytoclasts and amorphous organic matter (AOM) which together
comprise 45 to 90% of the palynofacies (Fig. 2). Phytoclasts are dominant in the bioturbated samples in the



lower part of the section, while AOM dominates the palynofacies in the laminated intervals. Other terrestrially
derived organic (wood fragments and charcoal) particles make up a small portion (<20%) of the palynofacies
and are not influenced by the alternating anoxic/oxic depositional conditions (Fig. 2). Palynomorphs were
divided into three groups: pollen, spores and aquatic palynomorphs. Pollen abundance relative to other
palynofacies is higher in the upper 0.5 m of the Schuchert Dal Formation and lowest metre of the Wordie Creek
Formation (5-20%), which is mainly caused by the high number of pollen fragments (sacci) during and
following the extinction event (Fig. 2). Spores have highest abundance in the upper metre of the Schuchert Dal
Formation (up to 35%), where also the spore/pollen ratio is highest. Aquatic palynomorphs (acritarchs and other
algal remains like *Tasmanites* and *Cymatiosphaera*) make up a very small fraction of the palynofacies in the
Schuchert Dal Formation (<2%), but become relatively more abundant in the Wordie Creek Formation (up to

160 14%).


**4.2 Aquatic palynomorphs**
Although Permian acritarch diversity is considered to be low, an overview of Permian phytoplankton by Lei et
al. (2013b) showed a richness of about 20-30 genera during most of the Permian, with acritarchs belonging to
the genera *Micrhystridium* and *Veryhachium* being the most common (Lei et al., 2013b, 2013a; Sarjeant and
Stancliffe, 1994). Since cysts are generally small in size it is difficult to study them under light microscope and
identification at species level can be difficult (Sarjeant, 1970). Several revisions and simplifications have been
proposed for the genera (Lei et al., 2013a; Sarjeant and Stancliffe, 1994). We follow the simple classification
scheme proposed by Lei et al. (2013a), in which different species of acritarchs are grouped together, based on
geometrical shape of the acritarchs. Photos of selected aquatic palynomorphs are shown in the plate of Fig. 3.
Concentrations of aquatic palynomorphs were on average ca.4 times higher in the laminated rocks
compared to the bioturbated rocks (Fig. 4). Acritarch abundance also declines during the extinction event. The
majority (25 - 100%) of the acritarchs belong to the *Micrhystridium breve* Group (Fig. 4). This group includes
all acritarchs with a spherical central body and numerous processes. The *Veryhachium laidii* Group (acritarchs
with a rectangular central body shape)*, the V. trispinosum* Group (triangular shape)*, V. cylindricum* Group
(ellipsoidal shape) and *M. pentagonale* Group (pentagonal or hexagonal shape) were mostly found in
bioturbated intervals of the upper part of the Schuchert Dal Formation, especially just before and during the
extinction interval (Fig. 4). The leiospheres include all aquatic palynomorphs with a spherical shape and either a
smooth surface or with small ornamentation, but no processes. The highest abundances (up to 70%) of
leiospheres are found in the laminated sediments of the Wordie Creek Formation (Fig. 4). *Cymatiosphaera* are
rare in the record and found in both the oxidized siltstones of the Schuchert Dal Formation, but also in some
intervals in the laminated siltstones. *Tasmanites* were found only in one sample, also during the marine
extinction phase. The apparent relatively higher abundance of *Cymatiosphaera* and *Tasmanites* in the top of the
Schuchert Dal Formation is a consequence low total count of the aquatic palynomorphs during the extinction
event. The most diverse samples were found in the bioturbated rocks of the upper metre of the Schuchert Dal
Formation, an interval that partly overlaps with the extinction interval.



The *Micrhystridium* Group has a diverse morphology, and shows variation in vesicle size and number
of processes. Most of the measured specimens are very similar to the species *M. breve* as described by Jansonius
(1962) in both the number and shape of the processes. However, the size of the central body and the length of
the processes of many of the measured acritarchs fall outside of the defined size-range of *M. breve*. The
distributions of both body size and process length are unimodal (Fig. 5), which strongly suggests that the
measured specimens all belong to a single species of *Micrhystridium*. Body size is variable and in the two-metre
interval spanning the formation boundary, acritarchs in the laminated rocks are on average larger compared to
those in the bioturbated rocks (Fig. 6). Process length decreases gradually from the start of the extinction event
until its end, and then remains shorter through to the top of the studied section. Process length not only changes
in absolute terms, but also relatively to the size of the body.

**5. Discussion**
**5.1 The extinction event and onset of anoxia**
Consistent with the results of Looy et al. (2001), we find a strong increase in spores, relative to pollen,
in the upper Schuchert Dal Formation (Fig. 2). Since this peak in spores coincides with the interval of marine
ecosystem collapse defined by Twitchett et al. (2001), we define the period with highest spore/pollen ratios as
the extinction interval. Twitchett et al. (2001) found large changes in marine plankton abundance in their study
of the Fiskegrav section, with a bloom just before the disappearance of trace fossils, followed by almost
complete absence of acritarchs immediately after the collapse of the marine ecosystem. This phytoplankton
bloom is not obvious in our higher resolution data, and may simply be an artefact of the relatively low-
resolution sampling of Twitchett et al., (2001), but the number of acritarchs is indeed greatly reduced during the
biotic crisis (Fig. 4). The peak in spore/pollen and decrease in acritarch abundance occurs in an interval of ca.
0.8m, which is comparable to the interval found by Twitchett et al., (2001) for marine and terrestrial ecosystem
collapse. Twitchett et al., (2001) estimated the collapse to have taken between 10 and 60 kyrs, however, recently
improved estimations of sedimentation rates in the Fiskegrav section (Mettam et al 2017) now give an estimated
duration of between 1.6 to 8 kyrs for the marine and terrestrial ecosystem collapse.
During the start of the biotic crisis rocks remain bioturbated but within the upper 0.5 m of the
Schuchert Dal Formation there are intervals with laminated rocks (Fig. 2), indicative for anoxia (Twitchett et al.,
2001). AOM, which is structureless organic matter, is dominant in the laminated rocks. AOM can derive from
degraded macrophyte, phytoplankton or higher plant tissue, or it may also be bacterially derived (Tyson, 1995).
Anoxic conditions can stimulate the production of marine AOM (Pacton et al., 2011), but also result in better
preservation after sedimentation (Tyson, 1995). The relative proportions of AOM versus phytoclasts thus
indicate the amount of oxygenation of the depositional environment (Tyson, 1995). A recent, high-resolution
study of the $\delta^{13}$C of total organic carbon (TOC) by Mettam et al., (2017) shows a negative shift of 5-6‰ that
begins 1.80m below the top of the Schuchert Dal Formation at Fiskegrav, (Fig. 2). The onset of this negative
excursion coincides closely with the last occurrence of Permian macroinvertebrate fossils (Mettam et al., 2017),
but pre-dates the collapse of marine and terrestrial ecosystems defined by disappearance of bioturbation.



Mettam et al. (2017) record a small, but consistent, offset in $\delta^{13}C$ values between the laminated/bioturbated
intervals (Fig. 2). This offset can be explained by the differences in organic matter source, which is mostly
phytoclasts (terrestrial) in the bioturbated intervals while AOM (marine and terrestrial) dominates in the
laminated intervals. However, the main shift is not associated with major changes in the palynofacies, and
occurs in a part of the section where the palynofacies mainly consist of terrestrially derived organic matter (Fig.
2). This negative shift in carbon isotopes associated with the late Permian extinction event is recognized
worldwide (e.g. Korte and Kozur, 2010) and usually precedes, the extinction event (e.g. Burgess and Bowring,
2015; Korte and Kozur, 2010). It has been interpreted as resulting from atmospheric changes in $\delta^{13}C$ values,
which can be directly, or indirectly, related to Siberian trap volcanism (e.g. Cui and Kump, 2015; Svensen et al.,

233    2009).

Mettam et al. (2017) studied redox conditions in the same section and showed that conditions became
potentially anoxic, and ferruginous ($Fe^{2+}$- rich) in the laminated intervals near the formation boundary. Within
the lowermost Wordie Creek Formation, from 0.6-0.7 m above the formation boundary, anoxic conditions are
confirmed by Fe speciation (Mettam et al., 2017). Since the extinction started before conditions became
ferruginous, spreading anoxia cannot be the main cause for the marine biotic crisis in the Fiskegrav section.
Grasby et al. (2015) distinguished three phases of extinction in the shallow marine sequence of Festningen
(Svalbard): first is the loss of carbonate macrofauna, followed by loss of siliceous sponges, and finally a loss of
all trace fossils. Thus, similar to what is observed in East Greenland, the first laminated sediments do not
represent the start of the biotic crisis. This is unsurprising as expanding marine hypoxia is a predicted
consequence of global warming, which was probably caused by elevated $CO_2$ flux from Siberian Trap
volcanism (Benton and Twitchett, 2003; Grard et al., 2005). Therefore, extinctions that occurred prior to the
appearance of widespread anoxia were more likely due to the direct, short-term effects of volcanic activity (e.g.
metal toxicity), or the more immediate effects of $CO_2$ rise or temperature increase. The laminated rocks are
associated with higher amount of pollen fragments (Fig. 2), which could be an indication for elevated terrestrial
organic matter input (e.g. soil erosion), while the higher amount of AOM can be partly explained by terrestrial
organic matter input, and partly by increased marine productivity and better preservation of organic matter due
to low oxygen conditions. Increased weathering and run-off are expected consequences of atmospheric and
hydrological changes associated with global warming. Algeo and Twitchett, (2010) demonstrated that sediment
accumulation rates greatly increased in shallow shelf seas in the latest Permian and earliest Triassic, consistent
with enhanced weathering and run-off, and Sephton et al. (2005) found evidence for large-scale soil erosion.

## 5.2 Sea-level and salinity changes

If the laminated rocks are indeed a consequence of enhanced runoff, this is expected to affect also
marine environmental conditions, such as a decrease in (surface) salinity. At the same time, sea-level
fluctuations at and near the boundary need to be considered, since these would also affect marine environmental
conditions and could potentially affect salinity. The Fiskegrav section has well preserved aquatic palynomorphs,
and their diversity, distribution and morphology can provide valuable information on marine environmental
changes. Palaeozoic studies have shown that acritarch diversity is generally higher in deeper, more distal





settings (e.g. Lei et al., 2012; Stricanne et al., 2004). In addition, different studies have shown that *Veryhachium*
favoured more open marine settings, while *Micrhystridium* favoured near-shore environments (e.g. Wall 1965,
Lei et al., 2012). It is possible that some of the leiospheres in the Fiskegrav section belong to the genus
*Leiosphaeridia,* as the dimensions are comparable to *L. microgranifera* and *L. minutissima* that were found in
Chinese Permian-Triassic sections (Lei et al., 2012), but the small ornamentations on both of those species is
different (Fig. 3). In these Chinese Permian sections the *Leiosphaeridia* are associated with deeper/more open
marine waters (Lei et al., 2012). On the other hand, studies of early Palaeozoic acritarchs show that, in general,
leiospheres, or sphaeromorphs, are most frequent in proximal environments (Li et al., 2004; Stricanne et al.,
2004). The leiosphere-group might be polyphyletic and some species might have prasinophyte affinities
(Colbath and Grenfell, 1995), which complicates their use for environmental reconstructions. Whatever the
biological origin of the leiosphere is, in this study site they apparently favoured the palaeoenvironmental
conditions associated with the deposition of the laminated rocks, starting during the extinction event.

274         In the Fiskegrav section, the decline in diversity from the upper Schuchert Dal Formation to the Wordie

Creek formation, together with the change in assemblage from *Veryhachium/Micrhystridium* to
*Micrhystridium/*leiosphere dominance could thus be interpreted as a change from a distal to a more proximal
setting, which would suggest a marine regression. Although this was once considered as a possible cause of the
late Permian extinction event (Hallam and Cohen, 1989), the current consensus is that eustatic sea-level fall
occurred prior to collapse of marine ecosystems and the extinction, and the subsequent transition from the
Permian to the Triassic happened during sea-level rise (Wignall and Hallam, 1992).  In some East Greenland
locations, apparently missing biozones and local erosive conglomeratic channels have been proposed as
evidence for sea-level fall near the boundary between the Schuchert Dal and Wordie Creek formations (Surlyk
et al., 1984), but Wignall and Twitchett (2002) demonstrated that in central Jameson Land the sedimentological
changes are consistent with sea-level rise, and that the erosive, conglomeratic channels occur at multiple
stratigraphic levels and postdate the extinction and Permian/Triassic boundary.

286         Thus, during the extinction event the acritarch assemblage changes to a more typical near-shore

assemblage, despite the ongoing sea-level rise. Instead of a marine regression, the observed shift in acritarch
assemblages could be explained by an increase in runoff which affects the near-shore environment by lowering
surface water salinity and delivering nutrients, both of which are found to affect the distribution of modern
phytoplankton (e.g. Bouimetarhan et al., 2009; Devillers and de Vernal, 2000; Pospelova et al., 2004; Zonneveld
et al., 2013). Dinocyst morphology has also been linked to environmental conditions (e.g., Mertens et al., 2009,
2011), with salinity identified as an important factor influencing process length (Ellegaard et al., 2002; Mertens
et al., 2011). Similarly, studies of acritarch process length and palaeoenvironment show that species and
individuals with longer processes are generally found in more offshore locations, while in inshore settings
acritarchs with shorter processes are more abundant (Lei et al., 2013a; Servais et al., 2004; Stricanne et al.,
2004). The average process length of specimens belonging to the *Micrhystridium breve* Group, show a
decreasing trend during the extinction interval, with higher values before the extinction event, and lower values
after the event (Fig. 6), which could thus be interpreted as a lowering in salinity during the extinction event.
Within the 2m interval including the extinction interval (-1 – +1m) there is also a consistent offset in body size
between samples from bioturbated and laminated rocks. The body size of some dinocysts has been found to vary
with temperature and salinity (e.g. Ellegaard et al., 2002; Mertens et al., 2012), however, the relation has not



been well studied. Possibly low oxygen or salinity conditions or increased nutrient availability influenced
acritarch body size in this interval. Runoff, lower salinity, and salinity stratification together can explain the
observed changes in palynofacies and acritarch records. The changed environmental conditions continued
through the aftermath of the biotic crisis, as shown by persistently low process lengths to the end of the study
interval (Fig. 6). Meanwhile, the ongoing marine transgression likely resulted in the shoreward migration of
oxygen deficient waters which explain the development of permanently anoxic conditions higher up in the
Wordie Creek Formation (over 0.6-0.7 m from the Fm boundary) (Mettam et al., 2017).
The development of widespread anoxia in the late Permian is usually explained by an expansion of
anoxic deep waters onto the shelf (e.g. Grasby and Beauchamp, 2009). However, increased runoff and the
development of estuarine circulation provide an alternative explanation for the development of anoxia in some
shallow marine environments. Furthermore, low salinity is known to negatively impact biomass and size of
modern marine invertebrates (e.g. Westerbom et al., 2002), and, along with higher temperatures and hypoxia,
might also be a contributing factor in the size reduction of marine organisms recorded in the aftermath of the
late Permian extinction event, such as observed for the bivalve *Claraia* the Permian-Triassic section of East-
Greenland (Twitchett, 2007). Low salinity conditions have previously been invoked to explain the dominance of
the brachiopod lingulid in post-extinction ecosystems (Zonneveld et al., 2007), because they are able to tolerate
a wide range of salinities as well as hypoxia (e.g. Hammen and Lum, 1977).  Lingulid fossils are found in many
Permian-Triassic (shallow) marine sections (e.g. Peng et al., 2007) and the lowering of salinities in shallow
marine settings and spreading anoxia provided ecological space for the lingulids (Peng et al., 2007; Rodland and
Bottjer, 2001). It has been suggested that rising sea water temperatures in the Tethys, resulted in strengthened
monsoonal activity (e.g. Winguth and Winguth, 2012) and enhanced precipitation and runoff in the areas
surrounding the Tethys (e.g. Parrish et al., 1982). Whether this would have had an effect on the study site is
questionable, because of its distance to the Tethys and relatively sheltered, inland position. Possibly river
systems existed which brought freshwater to the Greenland-Norway basin from the South. It is also possible that
rainfall increased locally. Mettam et al., (2017) speculated that changes in circulation patterns may have resulted
in onshore winds bringing moist air to the study site. In addition vegetation destruction and soil erosion may
also have contributed to enhanced runoff.

### 5.3 Sea water temperature

Besides salinity, increasing water temperatures can also affect process length in dinocysts (e.g. Mertens
et al., 2009, 2012) but this relationship is not always evident (e.g. Ellegaard et al., 2002). Sea water temperature
reconstructions for the late Permian and Early Triassic exist mainly from carbonate rich sections originating in
the Tethys Ocean (e.g. Joachimski et al., 2012; Schobben et al., 2014; Sun et al., 2012). These reconstructions
indicate increasing temperatures during the extinction event, and maximums temperatures are reached in the
Early Triassic (e.g. Joachimski et al., 2012; Kearsey et al., 2009; Schobben et al., 2014; Sun et al., 2012). Since
no water temperature reconstructions exist for the Greenland-Norway basin, at this time, it is not possible to
exclude the effects of rising water temperatures on the acritarch morphology. Increasing water temperatures
might be partly responsible for decreasing process length in acritarchs, however, a rising sea-level in





combination with rising temperatures are not able to explain the changes in the acritarch assemblages, which
indicate a shift from more open marine to near-shore conditions. Increased runoff cannot only explain the
reconstructed changes in palynofacies and acritarch records, but in addition, a reduction in salinity also has
significant implications for palaeotemperature reconstructions calculated from the oxygen isotopes of carbonates.
Most Permian-Triassic temperature records are from the Tethys or surrounding areas, where monsoon activity
has been proposed to increase (Winguth and Winguth, 2012) and thus salinity is expected to change as well.
Permian-Triassic oxygen isotope data for palaeotemperature estimates have been derived from brachiopod shells
(Kearsey et al., 2009) or conodont apatite (Joachimski et al., 2012; Schobben et al., 2014), with all studies
concluding that temperature rose through the extinction event and that Early Triassic seas were 'lethally hot'
(Sun et al., 2012). Oxygen isotope values partially reflect seawater salinity, although this is generally ignored in
palaeotemperature studies, because there is currently no independent proxy of salinity change, and so it is
assumed that salinity remained constant. If salinity was reduced in shelf seas due to hydrological changes (e.g.
enhanced precipitation and run-off), as it appears for East Greenland at least, then all of these studies may have
significantly over-estimated Early Triassic temperatures. The actual temperature rise associated with the
extinction event would have been lower and below 'lethal' levels, which is more consistent with fossil evidence
that the oceans were not completely devoid of life.

### 6. Conclusions

The highest resolution palynological sampling yet attempted for the Fiskegrav location of central Jameson Land
has shown significant, millennial-scale changes in marine and terrestrial environments.  The biotic crisis,
defined by a peak in the spore/pollen ratio which indicates widespread disruption and collapse of the terrestrial
flora, postdates the onset of the carbon-isotope excursion. The spore/pollen peak spans 0.8 m of section, which,
on current age estimates, gives an estimation of 1600 to 8000 years for the ecosystem collapse. This is much
shorter than previous estimations. The onset of the biotic crisis does not appear to coincide with any obvious
changes in lithology or sedimentary facies, and deposition of the shallow marine shelf sediments appears to have
taken place under oxygenated waters. During the latter stages of the biotic crisis, laminated beds begin to appear,
indicating periodic anoxic deposition. These laminated rocks contain palynological evidence for a more near-
shore setting, which contradicts the widely accepted evidence that globally sea-level was actually rising during
the extinction event. A preferred alternative explanation, consistent with palynological assemblages, is that
enhanced runoff, water column stratification and enhanced primary productivity led to the development of
anoxic bottom waters. Enhanced runoff would lead to elevated freshwater flux into shallow shelf seas, which
would have led to the development of a freshwater wedge, stratification and a reduction in salinity. Acritarch
process length is considered here to be a proxy for seawater salinity, and supports these inferred changes.
Increased runoff did not immediately lead to anoxia, possibly because precipitation gradually increased during
the extinction event until the freshwater flow was large enough to cause water column stratification. Increasing
sea water temperatures may have also affected acritarch morphology, and partially be responsible for the
reduced process length. However, rising temperatures alone cannot explain the reconstructed changes in the
palynofacies and acritarch records. In fact, sea water temperature reconstructions may overestimate the rise in



water temperatures during the extinction event, since possible changes in salinity are not included in the calculations.

**7. Data availability.** The data reported in this paper is available in tables in the supplementary files.

**Author contributions.** WMK designed the study. RJT provided material. EEvS analysed the samples for palynofacies and palynomorph content and performed measurements on acritarchs. EEvS wrote the paper with input from WMK and RJT.

**Competing interests.** The authors declare that they have no conflict of interest.

**Acknowledgements**

This work was funded by the Norwegian Research Council, project 234005, The Permian/Triassic evolution of the Timan-Pechora and Barents Sea Basins. We thank B.L. Berg and M.S. Naoroz for technical support. Samples were collected by RJT, who thanks G. Cuny and the Danish National Research Foundation for logistic and financial support.

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





**Figure 1.** Modern map of Jameson Land, East Greenland (adapted from Piasecki, 1984). The location of the studied section is indicated with a red circle. The right panel show the lithology of the Jameson Land section and sample depths. (* spore-peak after Looy et al., 2001, marine collapse after Twitchett et al., 2001).

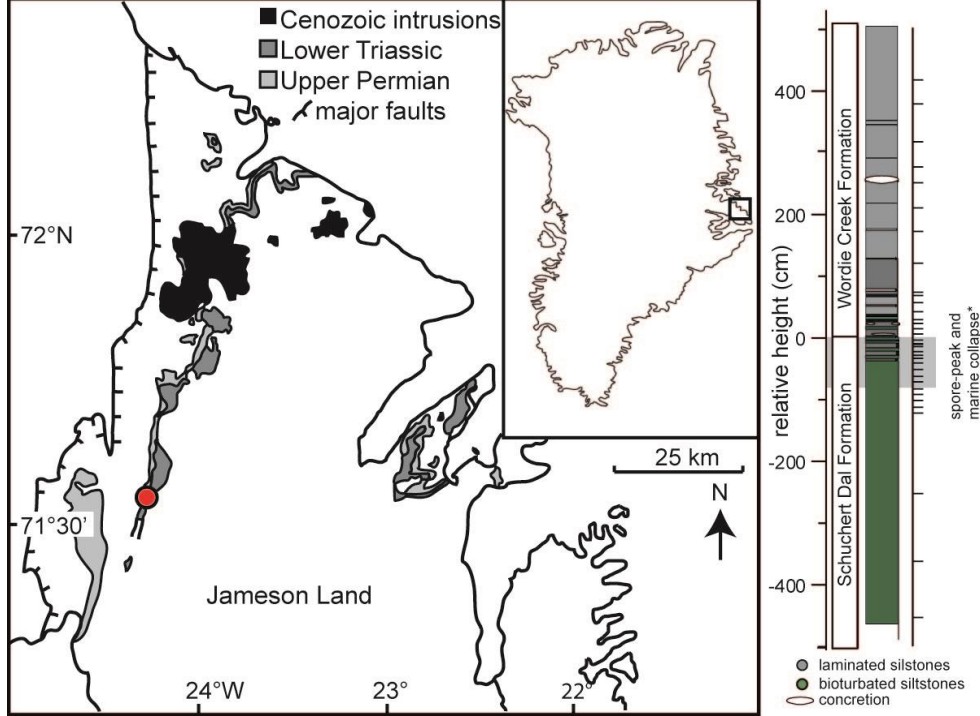





**Figure. 2**. Change in major groups of palynofacies. AOM= amorphous organic matter. The grey shaded box

highlights the biotic crisis as defined by the high spore/pollen ratios, which are indicative for the disappearance

of forest communities. Green shading indicates data from bioturbated rocks; grey shading indicates data from

laminated rocks. Grey horizontal bar indicates the extinction interval, based on high spore: pollen ratios.

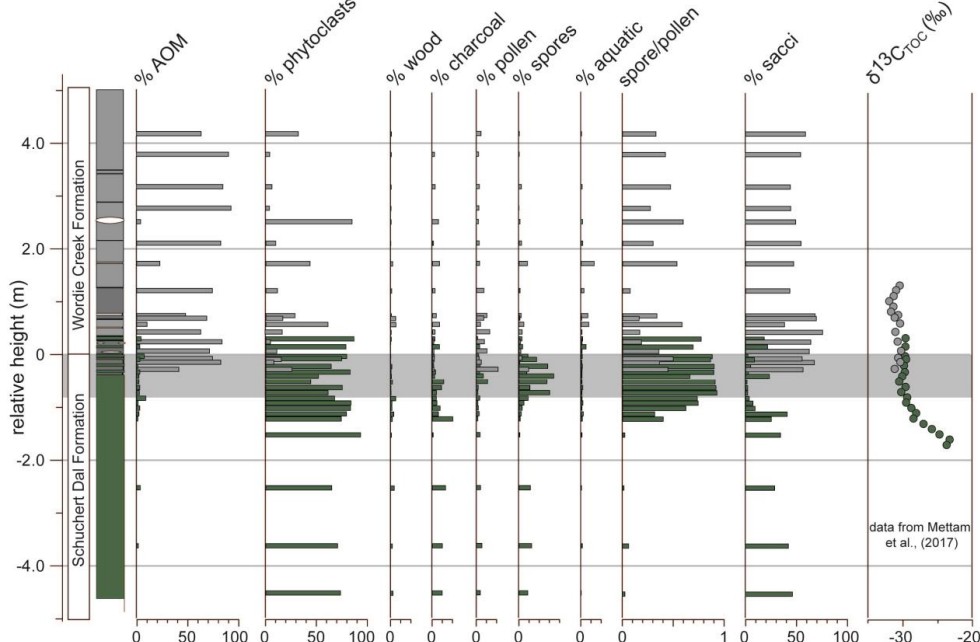



**Figure 3.** Aquatic palynomorphs plate. A) *M. pentagonale* – group, B) *V. laidii* - group  C) *Cymatiosphaera* sp.
D) and E) leiospheres F)-I) *M. breve* -group J) *M. pentagonale* - group K)-M) M. breve - group N)
*Cymatiosphaera* sp. O) and P) *M. breve* -group

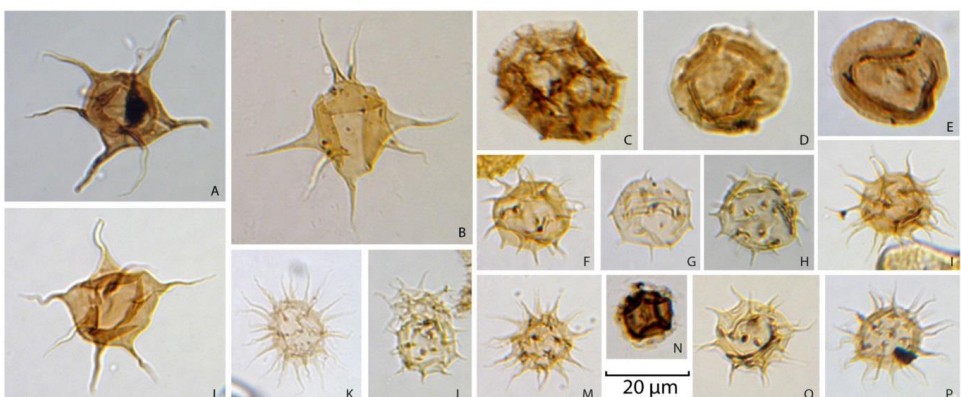








**Figure 4.** Aquatic palynomorphs. The figure showsconcentration of all aquatic palynomorphs andrelative
abundance specific acritarch and prasinophyte groups. Arrow indicate intervals with low counts (<20). Green
shading indicates data from bioturbated rocks; grey shading indicates data from laminated rocks. Grey
horizontal bar indicates the extinction interval, based on high spore: pollen ratios.

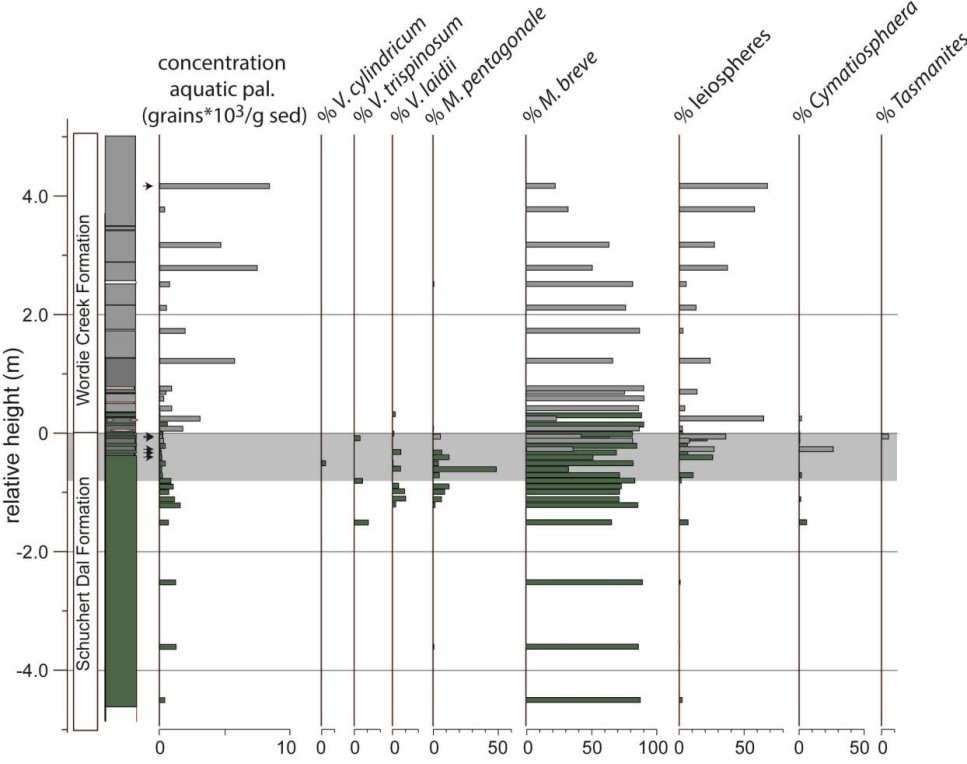





**Figure 5.** Size-frequency of body diamter (left) and proces length (right).

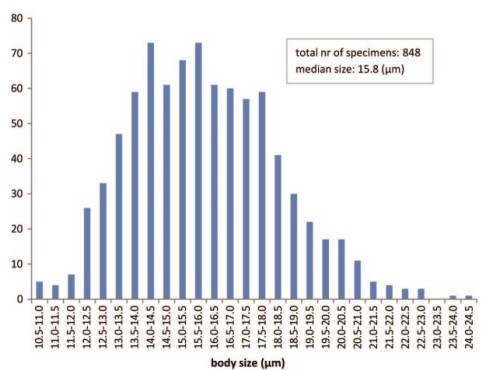
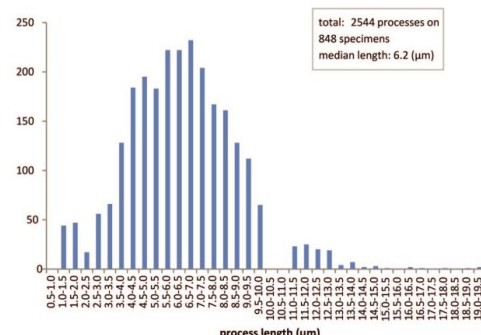






**Figure 6.** Changes in acritarch abundance and morphology. In the panel on the left are the main groups of
aquatic palynomorphs summarized. On the right: body size, process length, and process length relative to body
size. Where are low counts? Green shading indicates data from bioturbated rocks; grey shading indicates data
from laminated rocks. The grey bar indicates the extinction interval, based on high spore:pollen ratios. The five
small black arrows indicate in which samples the number of specimens used for measurements was low (4-16
specimens). Grey horizontal bar indicates the extinction interval, based on high spore: pollen ratios.

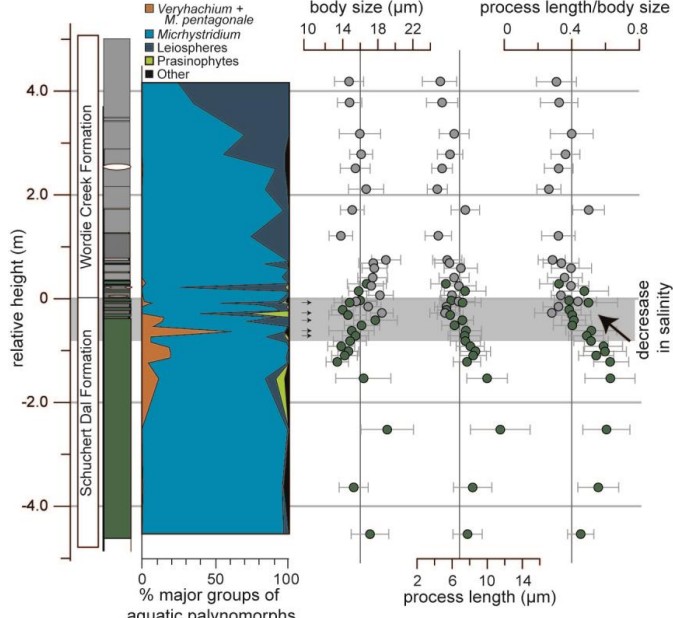