# Peer review of "Salinity changes and anoxia resulting from enhanced runoff during the late Permian global warming and mass extinction event"

_Climate of the Past, 2017_

## Referee Comment (RC1) · Anonymous Referee #1 · 18 Dec 2017

This paper documents nicely the potential of the process length of acritarchs as a palaeo-proxy for salinity.

After a first version of this manuscript, submitted to another journal, where the main problem was an over-interpretation of the results, the present version clearly points out the potential of morphology studies of organic-walled microphytoplankton for tracking ancient salinity of temperature changes.

This is still very much hypothetical, but the study is been carried out with precision, and the results show a clear trend. It is made clear that the interpretations have to take with caution. This study should be published; these results will surely stimulate further

investigations!

---

## Referee Comment (RC2) · Anonymous Referee #2 · 20 Dec 2017

General comments and specific issues

This is an interesting study that discusses acritarch processes length as new proxy for salinity changes in correspondence to the end-Permian mass extinction. The data are novel and interesting for a wide range of scientists working on palaeoenvironments reconstruction and the end-Permian mass extinction. The manuscript is well written and structured. However, given the novelty of the proxy I think the description and discussion of the used method require a more in-depth treatment, especially from a statistical point of view. Moreover, I think the main conclusion of the manuscript, i.e. salinity changes at the P/T boundary, is not sufficiently supported by the data, which

are in my opinion over interpreted. I therefore suggest major revision.

Main comments:

1) To study the changes in processes length across the section the authors measured three processes per specimen (Lines 140 – 141). The specimens selected for these study include "only those with a spherical central body and many (>10) simple processes with closed tips" belonging to the Micrhystridium breve Group (lines 137 – 138). Why did the author measure only 3 processes per specimen? How did they select the three processes to measure? In Fig. 3 it seems that the processes of M. breve Group acritarchs vary in length within a single specimen. What is the process length variability in one specimen? How could the selection of only 3 out of 10 processes bias the analysis? These methodological issues must be assessed, discussed and taken into account for the interpretation of the data. I strongly recommend the authors to expand the discussion on the methodological aspect, which I believe is the most important theme of this study.

2) The observed changes in acritarch average processes length are interpreted as the evidence of changing salinity in the water column. This conclusion seems to be based on the assumption that the Micrhystridium acritarchs considered for this study belong to one species. According to the authors, this assumption is suggested by the unimodal distribution of body size and processes length (Fig. 5). However, considering data through time (Fig. 6), there is indeed a difference (lines 192 – 194) in processes length and in body size between the acritarchs extracted from laminated rocks (after the extinction; Wordie Creek Formation) and those from bioturbated rocks (before the extinction; Schuchert Dal Formation). This is actually the main conclusion of the study. So, could these facies-dependant differences in both processes length and body size just mean that different species (with a different body size and length of the processes) are found in the different formations? Do the acritarchs' morphological differences represent a salinity-related mutation in one species through time or do these morphological characters show different Micrhystridium species? Acritarch concentrations at Fiskegrav, before and after the extinction, also suggest a change in the community structure, and in fact a change in the marine palynomorphs' assemblage is detected (see below, point 3). Similarly and as an example, van de Schootbrugge et al. (2007; P3, 244, 126–141) found an increase in actritarch abundance and diversity at the Triassic – Jurassic boundary.

3) The authors say that "studies of acritarch process length and palaeoenvironment show that species and individuals with longer processes are generally found in more offshore locations, while in inshore settings acritarchs with shorter processes are more abundant" (lines 293 – 295). These differences have been tentatively attributed to salinity differences between off-shore and nearshore environments (Servais et al., 2004; Palaeontology, 47, 395–414). So, could the change in average process length found in the studied section just be an effect of transport of acritarchs living in different environments? Is it possible that inshore species of Micrhystridium acritarchs with shorter process length, were transported to the more distal setting? The processes length change is coupled to a change in the assemblage from "Veryhachium/Micrhystridium to Micrhystridium/leiosphere dominance" (lines 275 – 276). Leiosphere are common in fluvial-deltaic environments (e.g. Zavattieri & Pramparo 2006; Palaeontology, 49, 1185–1209), and indeed in the Fiskegrav section the reduction of average process length is coupled to an increase in leiosphere abundance (Fig. 6). The authors indeed acknowledge that "during the extinction event the acritarch assemblage changes to a more typical near-shore assemblage, despite the ongoing sea-level rise" (lines 286 – 287), but they do not discuss the possible effect of transport. This is a crucial point that must be discussed to understand whether the observed changes in acritarch morphology reflect actual changes in salinity of the water column or just transport of nearshore species towards more distal environments after the extinction interval.

4) Local vs global signal. Data collected in this study come from one stratigraphic section that has been deposited in a narrow basin. The observed changes in acritarch assemblage, body size, and process length thus represent a very local signal. What is

the significance of this local change in a global perspective?

5) I think the title of the manuscript does not satisfactorily mirror its content.

Other minor comments

Line 67: Servais et al. (2004) talk about Cambrian – Ordovician acritarchs. Given the fact that acritarchs are a group of uncertain affinity, is there any more appropriate citation supporting the statement that "many acritarchs are, however, found exclusively in marine rocks"? In other words, is this true also for the Permian – Triassic?

Lines 72 – 75: The reference provided (Martens et al., 2009) discusses the effect of salinity in one modern dinocyst morphology. Is it possible to provide any reference about the effects of salinity on acritarchs?

Lines 192 – 194: The numbers, e.g. average body size, average processes length, errors, etc…, to support these statements should be presented. This would help the reader. A simple reference to the figure is not sufficient.

Is it possible to introduce in the results the changes in acritarch assemblage later discussed in 286 – 287 and following lines? How could this change affect the measurement and interpretation of acritarch processes length?

---

## Author Comment (AC1) · 18 Jan 2018

We thank the reviewer for the positive feedback and we hope that this study will indeed stimulate further studies on acritarch morphology.
* * *

---

## Author Comment (AC2) · 18 Jan 2018

We like to thank the reviewer for the constructive comments. Below we provide a point-by-point reply to concerns prompted by the reviewer.

Comment #1) To study the changes in processes length across the section the authors measured three processes per specimen (Lines 140 – 141). The specimens selected for these study include "only those with a spherical central body and many (>10) simple processes with closed tips" belonging to the Micrhystridium breve Group (lines 137 – 138). Why did the author measure only 3 processes per specimen? How did they select the three processes to measure? In Fig. 3 it seems that the processes of M. breve

Group acritarchs vary in length within a single specimen. What is the process length variability in one specimen? How could the selection of only 3 out of 10 processes bias the analysis? These methodological issues must be assessed, discussed and taken into account for the interpretation of the data. I strongly recommend the authors to expand the discussion on the methodological aspect, which I believe is the most important theme of this study.

Reply #1) The selection of the processes on the acritarchs was limited by the fact that the acritarchs are compressed into a two dimensional shape and, as a consequence, some of the processes are not visible. We therefore chose a focal plane that showed the most, and best preserved processes. From these we measured the three longest, following the methodology by Mertens et al., (2012). We will clarify this in the methodology.

We agree with the reviewer that the length of the processes varies within a single specimen. In about 50% of the measured acritarchs this variation within one specimen is smaller than 1 $\mu$m, in 28% the variation within one specimen was between 1-2$\mu$m, in 15% between 2-3 $\mu$m, and only in 7% larger than 3 $\mu$m. In addition, we measured 20-30 specimens per sample in order to minimize noise and decrease the standard error within one sample. The overall trend over the studied section is a shortening of process length of 3-5 $\mu$m. The variability of the process length within one specimen therefore is smaller than the general trend. We expanded the method section to explain better the selection of the processes, and in the results section we now describe the degree of variability within individual acritarchs.

Comment #2) The observed changes in acritarch average processes length are interpreted as the evidence of changing salinity in the water column. This conclusion seems to be based on the assumption that the Micrhystridium acritarchs considered for this study belong to one species. According to the authors, this assumption is suggested by the unimodal distribution of body size and processes length (Fig. 5). However, considering data through time (Fig. 6), there is indeed a difference (lines 192 – 194)

in processes length and in body size between the acritarchs extracted from laminated rocks (after the extinction; Wordie Creek Formation) and those from bioturbated rocks (before the extinction; Schuchert Dal Formation). This is actually the main conclusion of the study. So, could these facies-dependant differences in both processes length and body size just mean that different species (with a different body size and length of the processes) are found in the different formations? Do the acritarchs' morphological differences represent a salinity-related mutation in one species through time or do these morphological characters show different Micrhystridium species? Acritarch con centrations at Fiskegrav, before and after the extinction, also suggest a change in the community structure, and in fact a change in the marine palynomorphs' assemblage is detected (see below, point 3). Similarly and as an example, van de Schootbrugge fet al. (2007; P3, 244, 126–141) found an increase in actritarch abundance and diversity at the Triassic – Jurassic boundary. Reply:

Reply #2) Although several different Micrhystridium-species have been described for the late Permian (e.g. Lei et al., 2013b), it has also been noted by several authors that the different species are difficult to distinguish (Lei et al., 2013a; Sarjeant and Stancliffe, 1994). We follow the simple classification proposed by Lei et al., (2013) in which all spherical specimens belonging to the Micrhystridium/Veryhachium complex are grouped together in a Micrhystridium breve-group (independent of the process length or body size). The purpose of our study is to find out if some of the observed variability in morphology of the late Permian Micrhystridium-group might be explained by environmental conditions rather than different species. Thus, we indeed assume that the acritarchs of the Micrhystridium breve-group, found in our section, belong to one species. This assumption is tested by the frequency analysis of all body size and process length. The unimodal distribution (uniform morphology) of the studied acritarch population does not support the presence of 2 or more different species.

We agree with the reviewer, that there is indeed, on average, a difference in process length between acritarchs from the laminated and form the bioturbated intervals. However, while the changes in facies are abrupt, the change in process length is gradual, and starts before the facies change. Also in body size there is no consistent off-set throughout the whole record. So, both the changes observed in process length and body size are not directly related to facies.

Comment #3) The authors say that "studies of acritarch process length and palaeoenvironment show that species and individuals with longer processes are generally found in more offshore locations, while in inshore settings acritarchs with shorter processes are more abundant" (lines 293 – 295). These differences have been tentatively attributed to salinity differences between off-shore and nearshore environments (Servais et al., 2004; Palaeontology, 47, 395–414). So, could the change in average process length found in the studied section just be an effect of transport of acritarchs living in different environments? Is it possible that inshore species of Micrhystridium acritarchs with shorter process length, were transported to the more distal setting? The processes length change is coupled to a change in the assemblage from "Veryhachium/Micrhystridium to Micrhystridium/leiosphere dominance" (lines 275 – 276). Leiosphere are common in fluvial-deltaic environments (e.g. Zavattieri & Pramparo 2006; Palaeontology, 49, 1185–1209), and indeed in the Fiskegrav section the reduction of average process length is coupled to an increase in leiosphere abundance (Fig. 6). The authors indeed acknowledge that "during the extinction event the acritarch assemblage changes to a more typical near-shore assemblage, despite the ongoing sea-level rise" (lines 286 – 287), but they do not discuss the possible effect of transport. This is a crucial point that must be discussed to understand whether the observed changes in acritarch morphology reflect actual changes in salinity of the water column or just transport of nearshore species towards more distal environments after the extinction interval.

Reply #3) Transport is rejected as an explanation of changing process length for two reasons: First, the preservation is very good throughout, and especially in the laminated rocks the Micrhystridium are of good quality, which argues against large-scale

transport of the acritarchs. Second, and most importantly, if there was transport of acritarchs with short processes into a deeper water environment, this would result in a mixture of acritarchs with long and short processes in those samples. This is not the case. Although there is certainly variability within each sample in process length, the samples do not show such a bimodal distribution. A figure showing process length and body size for each specimen, and average values per sample, will be added to the supplementary file.

Comment #4) Local vs global signal. Data collected in this study come from one stratigraphic section that has been deposited in a narrow basin. The observed changes in acritarch assemblage, body size, and process length thus represent a very local signal. What is the significance of this local change in a global perspective?

Reply #4) The data are indeed local, but the inferred changes in rainfall suggest changes in the hydrological cycle and support climate models for the later Permian. Future research on acritarch process length from other locations will provide further insight into climate and oceanographic changes in the later Permian, and will allow predictions made from global climate models to be tested.

Comment #5) I think the title of the manuscript does not satisfactorily mirror its content.

Reply #5): We kindly disagree with the reviewer and think the title describes the content well.

Other minor comments

Comment)Line 67: Servais et al. (2004) talk about Cambrian – Ordovician acritarchs. Given the fact that acritarchs are a group of uncertain affinity, is there any more appropriate citation supporting the statement that "many acritarchs are, however, found exclusively in marine rocks"? In other words, is this true also for the Permian – Triassic?

Reply) We change the sentence to "considered to be phytoplankton" and add a citation of Lei et al., 2013 for a review of late Permian acritarchs found in marine successions.

Comment) Lines 72 – 75: The reference provided (Martens et al., 2009) discusses the effect of salinity in one modern dinocyst morphology. Is it possible to provide any reference about the effects of salinity on acritarchs?

Reply) We added a reference to Servais et al., 2004, Palaeontology, Vol. 47, Part 2. In which the possible effects of salinity changes on acritarch morphology of Cambrian-Ordovician acritarchs is discussed.

Comment) Lines 192 – 194: The numbers, e.g. average body size, average processes length, errors, etc. . ., to support these statements should be presented. This would help the reader. A simple reference to the figure is not sufficient. Is it possible to introduce in the results the changes in acritarch assemblage later discussed in 286 – 287 and following lines? How could this change affect the measurement and interpretation of acritarch processes length?

Reply) Following the advice of the reviewer, we added values to the last paragraph in results section 4.2 aquatic palynomorphs. The text now introduces better the discussion part of section 5.2. The values are also given in a table in the supplementary files. As discussed above, changes in assemblage do not affect the measurement and interpretation of acritarch process length.

References used in the response:

Lei, Y., Servais, T., Feng, Q. and He, W.: Latest Permian acritarchs from South China and the Micrhystridium/Veryhachium complex revisited, Palynology, 37(2), 325–344, doi:10.1080/01916122.2013.793625, 2013a.

Lei, Y., Servais, T. and Feng, Q.: The diversity of the Permian phytoplankton, Rev. Palaeobot. Palyno., 198, 145–161, doi:10.1016/j.revpalbo.2013.03.004, 2013b.

Mertens, K. N., Bringué, M., Van Nieuwenhove, N., Takano, Y., Pospelova, V., Rochon, A., De Vernal, A., Radi, T., Dale, B., Patterson, R. T., Weckström, K., Andrén, E., Louwye, S. and Matsuoka, K.: Process length variation of the cyst of the dinoflagellate

Protoceratium reticulatum in the North Pacific and Baltic-Skagerrak region: calibration as an annual density proxy and first evidence of pseudo-cryptic speciation, J. Quaternary Sci., 27(7), 734–744, doi:10.1002/jqs.2564, 2012.

Sarjeant, W. A. S. and Stancliffe, R. P. W.: The Micrhystridium and Veryhachium Complexes (Acritarcha: Acanthomorphitae and Polygonomorphitae): A Taxonomic Reconsideration, Micropaleontology, 40(1), 1–77, doi:10.2307/1485800, 1994.

[Figure]

**Fig. 1.** Additional figure for suppl. showing all data points

---

## Author Response (AR1)

We thank the two reviewers for providing comments on our manuscript. The reviewers' original comments are shown in black with our responses in red. Also in the manuscript attached, changes made to the earlier version are given in red.

In response to comments by reviewer #2, we calculated variability in process length within each specimen, and found these in general to be small (a table was added to the supplementary file to show the variability and the frequency with which it occurs). We also clarified better the methodology and discussed other options for process length variability. We also added a figure to the supplementary file showing the individual measurements per acritarch and the averages for each interval (Fig. S1).

**Anonymous Reviewer #1**

This paper documents nicely the potential of the process length of acritarchs as a palaeo-proxy for salinity.
After a first version of this manuscript, submitted to another journal, where the main problem was an over-interpretation of the results, the present version clearly points out the potential of morphology studies of organic-walled microphytoplankton for tracking ancient salinity of temperature changes.
This is still very much hypothetical, but the study is been carried out with precision, and the results show a clear trend. It is made clear that the interpretations have to take with caution. This study should be published; these results will surely stimulate further investigations!

We like to thank the reviewer for the positive feedback. Feedback on the previous version of this manuscript, mentioned by the reviewer has been very helpful and improved the manuscript a great deal. We hope that this study will indeed stimulate further research on acritarch morphology.

**Anonymous Reviewer #2**

We like to thank the reviewer for the constructive comments. Below each comment we provide a response.

Comment:

1) To study the changes in processes length across the section the authors measured three processes per specimen (Lines 140 – 141). The specimens selected for these study include "only those with a spherical central body and many (>10) simple processes with closed tips" belonging to the Micrhystridium breve Group (lines 137 – 138). Why did the author measure only 3 processes per specimen? How did they select the three processes to measure? In Fig. 3 it seems that the processes of M. breve Group acritarchs vary in length within a single specimen. What is the process length variability in one specimen? How could the selection of only 3 out of 10 processes bias the analysis? These methodological issues must be assessed, discussed and taken into account for the interpretation of the data. I strongly recommend the authors to expand the discussion on the methodological aspect, which I believe is the most important theme of this study.

Reply:

The selection of the processes on the acritarchs was limited by the fact that the acritarchs are compressed into a two dimensional shape and, as a consequence, some of the processes are not visible. We therefore chose a focal plane that showed the most, and best preserved processes. From these we measured the three longest, following the methodology by Mertens et al., (2012). We will clarify this in the methodology.

We agree with the reviewer that the length of the processes varies within a single specimen. In about 50% of the measured acritarchs this variation within one specimen is smaller than 1 µm, in 28% the variation within one specimen was between 1-2µm, in 15% between 2-3 µm, and only in 7% larger than 3 µm. In addition, we measured 20-30 specimens per sample in order to minimize noise and decrease the standard error within one sample. The overall trend over the studied section is a shortening of process length of 3-5 µm. The variability of the process length within one specimen therefore is smaller than the general trend. We expanded the method section to explain better the selection of the processes, and in the results section we now describe the degree of variability within individual acritarchs.

2) The observed changes in acritarch average processes length are interpreted as the evidence of changing salinity in the water column. This conclusion seems to be based on the assumption that the Micrhystridium acritarchs considered for this study belong to one species. According to the authors, this assumption is suggested by the unimodal distribution of body size and processes length (Fig. 5). However, considering data through time (Fig. 6), there is indeed a difference (lines 192 – 194) in processes length and in body size between the acritarchs extracted from laminated rocks (after the extinction; Wordie Creek Formation) and those from bioturbated rocks (before the extinction; Schuchert Dal Formation). This is actually the main conclusion of the study. So, could these facies-dependant differences in both processes length and body size just mean that different species (with a different body size and length of the processes) are found in the different formations? Do the acritarchs' morphological differences represent a salinity-related mutation in one species through time or do these morphological characters show different Micrhystridium species? Acritarch con centrations at Fiskegrav, before and after the extinction, also suggest a change in the community structure, and in fact a change in the marine palynomorphs' assemblage is detected (see below, point 3). Similarly and as an example, van de Schootbrugge fet al. (2007; P3, 244, 126–141) found an increase in actritarch abundance and diversity at the Triassic – Jurassic boundary.

Reply:

Although several different Micrhystridium-species have been described for the late Permian (e.g. Lei et al., 2013b), it has also been noted by several authors that the different species are difficult to distinguish (Lei et al., 2013a; Sarjeant and Stancliffe, 1994). We follow the simple classification proposed by Lei et al., (2013) in which all spherical specimens belonging to the Micrhystridium/Veryhachium complex are grouped together in a Micrhystridium breve-group (independent of the process length or body size). The purpose of our study is to find out if some of the observed variability in morphology of the late Permian Micrhystridium-group might be explained by environmental conditions rather than different species. Thus, we indeed assume that the acritarchs of the Micrhystridium breve-group, found in our section, belong to one species. This assumption is tested by the frequency analysis of all body size and process length. The unimodal distribution (uniform morphology) of the studied acritarch population does not support the presence of 2 or more different species.

We agree with the reviewer, that there is indeed, on average, a difference in process length between acritarchs from the laminated and form the bioturbated intervals. However, while the changes in facies are abrupt, the change in process length is gradual, and starts before the facies change. Also in body size there is no consistent off-set throughout the whole record. So, both the changes observed in process length and body size are not directly related to facies.

3) The authors say that "studies of acritarch process length and palaeoenvironment show that species and individuals with longer processes are generally found in more offshore locations, while in inshore settings acritarchs with shorter processes are more abundant" (lines 293 – 295). These differences have been tentatively attributed to salinity differences between off-shore and nearshore environments (Servais et al., 2004; Palaeontology, 47, 395–414). So, could the change in average process length found in the studied section just be an effect of transport of acritarchs living in different environments? Is it possible that inshore species of Micrhystridium acritarchs with shorter process length, were transported to the more distal setting? The processes length change is coupled to a change in the assemblage from "Veryhachium/Micrhystridium to Micrhystridium/leiosphere dominance" (lines 275 – 276). Leiosphere are common in fluvial-deltaic environments (e.g. Zavattieri & Pramparo 2006; Palaeontology, 49, 1185–1209), and indeed in the Fiskegrav section the reduction of average process length is coupled to an increase in leiosphere abundance (Fig. 6). The authors indeed acknowledge that "during the extinction event the acritarch assemblage changes to a more typical near-shore assemblage, despite the ongoing sea-level rise" (lines 286 – 287), but they do not discuss the possible effect of transport. This is a crucial point that must be discussed to understand whether the observed changes in acritarch morphology reflect actual changes in salinity of the water column or just transport of nearshore species towards more distal environments after the extinction interval.

Reply: Transport is rejected as an explanation of changing process length for two reasons: First, the preservation is very good throughout, and especially in the laminated rocks the Micrhystridium are of good quality, which argues against large-scale transport of the acritarchs. Second, and most importantly, if there was transport of acritarchs with short processes into a deeper water environment, this would result in a mixture of acritarchs with long and short processes in those samples. This is not the case. Although there is certainly variability within each sample in process length, the samples do not show such a bimodal distribution. A figure showing process length and body size for each specimen, and average values per sample, will be added to the supplementary file.

4) Local vs global signal. Data collected in this study come from one stratigraphic section that has been deposited in a narrow basin. The observed changes in acritarch assemblage, body size, and process length thus represent a very local signal. What is the significance of this local change in a global perspective?

Reply: The data are indeed local, but the inferred changes in rainfall suggest changes in the hydrological cycle and support climate models for the later Permian. Future research on acritarch process length from other locations will provide further insight into climate and oceanographic changes in the later Permian, and will allow predictions made from global climate models to be tested.

5) I think the title of the manuscript does not satisfactorily mirror its content.

Reply: We kindly disagree with the reviewer and think the title describes the content well.

Other minor comments

Line 67: Servais et al. (2004) talk about Cambrian – Ordovician acritarchs. Given the fact that acritarchs are a group of uncertain affinity, is there any more appropriate citation supporting the statement that "many acritarchs are, however, found exclusively in marine rocks"? In other words, is this true also for the Permian – Triassic?

Reply: We change the sentence to "considered to be phytoplankton" and add a citation of Lei et al., 2013 for a review of late Permian acritarchs found in marine successions.

Lines 72 – 75: The reference provided (Martens et al., 2009) discusses the effect of salinity in one modern dinocyst morphology. Is it possible to provide any reference about the effects of salinity on acritarchs?

Reply: We added a reference to Servais et al., 2004, Palaeontology, Vol. 47, Part 2. In which the possible effects of salinity changes on acritarch morphology of Cambrian-Ordovician acritarchs is discussed.

Lines 192 – 194: The numbers, e.g. average body size, average processes length, errors, etc. . ., to support these statements should be presented. This would help the reader. A simple reference to the figure is not sufficient. Is it possible to introduce in the results the changes in acritarch assemblage later discussed in 286 – 287 and following lines? How could this change affect the measurement and interpretation of acritarch processes length?

Reply: Following the advice of the reviewer, we added values to the last paragraph in results section 4.2 aquatic palynomorphs. The text now introduces better the discussion part of section 5.2. The values are also given in a table in the supplementary files. As discussed above, changes in assemblage do not affect the measurement and interpretation of acritarch process length.

[revised manuscript text omitted]